# The Emergence of TRP Channels Interactome as a Potential Therapeutic Target in Pancreatic Ductal Adenocarcinoma

**DOI:** 10.3390/biomedicines11041164

**Published:** 2023-04-13

**Authors:** Yuanyuan Wei, Ahmad Taha Khalaf, Cao Rui, Samiah Yasmin Abdul Kadir, Jamaludin Zainol, Zahraa Oglah

**Affiliations:** 1Basic Medical College, Chengdu University, Chengdu 610106, China; 2Faculty of Medicine, Widad University College, BIM Point, Bandar Indera Mahkota, Kuantan 25200, Malaysia; 3School of Science, Auckland University of Technology (AUT), 55 Wellesley Street, Auckland 1010, New Zealand

**Keywords:** pancreatic cancer, ductal carcinoma, adenocarcinoma, tumor, TRP channel, pathogenesis

## Abstract

Integral membrane proteins, known as Transient Receptor Potential (TRP) channels, are cellular sensors for various physical and chemical stimuli in the nervous system, respiratory airways, colon, pancreas, bladder, skin, cardiovascular system, and eyes. TRP channels with nine subfamilies are classified by sequence similarity, resulting in this superfamily’s tremendous physiological functional diversity. Pancreatic Ductal Adenocarcinoma (PDAC) is the most common and aggressive form of pancreatic cancer. Moreover, the development of effective treatment methods for pancreatic cancer has been hindered by the lack of understanding of the pathogenesis, partly due to the difficulty in studying human tissue samples. However, scientific research on this topic has witnessed steady development in the past few years in understanding the molecular mechanisms that underlie TRP channel disturbance. This brief review summarizes current knowledge of the molecular role of TRP channels in the development and progression of pancreatic ductal carcinoma to identify potential therapeutic interventions.

## 1. Introduction

Pancreatic Ductal Adenocarcinoma (PDAC) is the most common and highly aggressive form of pancreatic cancer, with a low survival rate, making it one of the deadliest cancers [1]. The early phase of pancreatic cancer poses a challenge in diagnosis and treatment, resulting in a notably low survival rate. The 5-year survival rate for this type of cancer is only about 20% [2]. Early detection and treatment are essential, as the disease is often not diagnosed until it has spread to other body parts. Despite advances in medical treatments, the prognosis for pancreatic cancer remains poor, making research and efforts to improve early detection and treatment crucial [1,2,3].

Consequently, developing new treatment options is crucial to improve patient outcomes. Transient Receptor Potential (TRP) channels are a family of nonselective ion channels mediating various physiological functions in many cell types. These channels have recently been identified as critical regulators of pancreatic cancer cell proliferation, invasion, and metastasis [4,5,6,7]. Targeting TRP channels represents an exciting new strategy for inhibiting pancreatic cancer cell growth and survival. Because TRP channels also play essential roles in cellular signaling and allow the host cell to respond to benign or detrimental environmental changes, understanding how each TRP channel responds to its unique forms of activation energy is critical and crucial as its impairment may lead to several diseases, especially carcinogenesis.

Moreover, the development of effective treatment methods for pancreatic cancer has been hindered by the lack of understanding of the pathogenesis, partly due to the difficulty in studying human tissue samples. Despite this, scientific research on this topic has witnessed steady development in the past few years in understanding the molecular mechanisms that underlie TRP channel disturbance [4,5,6,7]. The focus is on exploring the pathogenesis of pancreatic cancer and its related pathways. Several studies have shown that TRP channels are frequently overexpressed in pancreatic cancer cells, and their expression levels correlate with poor prognosis and increased tumor growth. Transient receptor potential vanilloid 1 (TRPV1) is a calcium-permeable ion channel gated by the pungent constituent of red chili pepper, capsaicin, and related chemicals from the group of vanilloids, as well as by noxious heat stimuli [4,5,6,7]. TRPV1 has been linked to promoting PDAC cell proliferation and migration. This brief review summarizes current knowledge of the molecular role of TRP channels in the development and progression of pancreatic ductal carcinoma to identify potential therapeutic interventions.

## 2. Pancreatic Ductal Adenocarcinoma (PDAC)

PDAC is a malignant epithelial tumor originating from the pancreas’ ductal cells [8]. It is the most common type of pancreatic cancer and has a poor prognosis with a 5-year survival rate of less than 10% [8]. PDAC is a complex disease that arises from a series of genetic and epigenetic changes leading to the development of cancer cells. These changes may include mutations in key oncogenes, such as KRAS, TP53, and SMAD4, and loss of tumor suppressor genes, such as CDKN2A. In addition, risk factors such as smoking, obesity, diabetes, and exposure to certain chemicals and toxins have been implicated in developing PDAC, along with chronic inflammation and oxidative stress [8,9,10].

Diagnosis of PDAC is challenging due to the lack of specific symptoms in the early stages of the disease. Symptoms such as abdominal pain, weight loss, jaundice, and fatigue are often non-specific and may not be present until the disease has progressed [9,10,11]. Imaging techniques such as computed tomography (CT) and magnetic resonance imaging (MRI), can visualize the pancreas and detect abnormal growth. At the same time, endoscopic ultrasound (EUS) and biopsy can confirm the diagnosis [8,9,10,11].

Treatment of PDAC is multi-disciplinary and depends on the stage of the disease. Surgery is the preferred early-stage PDAC treatment to remove the tumor [8,9]. Oncologists employ chemotherapy and radiation therapy to reduce the size of cancerous tumors and inhibit their growth in patients with locally advanced and metastatic diseases. Novel therapies, such as immunotherapy and targeted therapy, are being investigated to improve outcomes for patients with PDAC [8,9,10,11], even though PDAC is a complex and aggressive cancer with a poor prognosis. Advances in understanding the genetic and epigenetic changes underlying PDAC have led to the development of new therapies. However, much work still needs to be conducted to improve outcomes for patients with this disease.

Nevertheless, recent studies have shown that the expression and function of specific transient receptor potential (TRP) channels, such as TRPV1 and TRPM8, are upregulated in PDAC cells, promoting cell growth and survival [11,12,13]. Targeting these TRP channels with specific inhibitors has induced apoptosis and inhibited cell proliferation in PDAC cells, suggesting a potential therapeutic benefit for patients with this disease [12,13]. Nonetheless, it is essential to note that this is a relatively new area of research. Further studies are needed to confirm the efficacy and safety of TRP channel inhibitors in treating PDAC. It is also crucial to consider the potential off-target effects of these inhibitors, as TRP channels are involved in many physiological processes, including sensory transduction and temperature regulation [12,13].

## 3. TRP Channels in Pancreatic Ductal Adenocarcinoma (PDAC)

Integral membrane proteins, or TRP channels, are cellular sensors for various physical and chemical stimuli in the nervous system, respiratory airways, colon, pancreas, bladder, skin, cardiovascular system, and eyes [14,15]. TRP channels are a class of cationic channels that are comparatively non-specific and are primarily found on the plasma membrane site of animal tissues. These channels react to a wide range of heterogeneous stimuli, such as free cytosolic Ca^2+^ ions, depletion of Ca^2+^ stores in the endoplasmic reticulum (ER), endogenous and exogenous chemical mediators, physical stimuli such as mechanical force (stretch-sensitive), temperature (thermo-sensitive), and many others. In addition, several TRP channels have unique characteristics, such as being ligand-gated and voltage-sensitive [14,15]. These TRP channels mediate several physiological and pathological processes including pain, temperature, pressure, vision, taste, inflammation, and pressure perception. As a result, the TRP ion channel family is considered polymodal with various properties. Several TRP ion channels mediate calcium influx into the cells [14,15,16].

Transient receptor potential channels (TRP channels) were first described in 1969 and subsequently named according to their electrophysiological function as transient receptor potential ion channels [15]. The TRP superfamily includes seven subfamilies based on sequence homology. These are TRPC (canonical), TRPV (vanilla), TRPM (melastatin), TRPP (polycystin), TRPML (mucolipin), TRPA (ankyrin), and one found only in invertebrates and fish, TRPN (NOMPC-like) [15]. TRP channels are essential for human health, as mutations in at least four channels result in morbidity and disease [14,15,16]. The first comprises the “classical” TRPs, the TRPC subfamily. The ducts of these families have a variety of functions, including the sensation of pain, temperature, taste, pressure, and vision [14,15,16]. Although research has identified the crucial role that TRP ion channels play in the development and progression of various types of cancer, translating this knowledge into effective practical and therapeutic applications has proven challenging. Studies have indicated a growing body of evidence linking TRP channels to the development of exocrine pancreatic cancer. The altered expression of various TRP proteins plays a critical role in tumor formation, proliferation, and migration [17,18]. TRP channel family members have also been reported as an excellent prognostic marker and a target for cancer drug therapy in recent decades [17,18].

The role of TRP channels in PDAC has been the subject of very recent studies, as summarized in the following Table 1:

Current therapeutic options for PDAC are minimal, and transient target receptor potential ankyrin 1 (TRPA1) is an attractive new therapeutic strategy [22,23,24,25,26,27,28]. TRPA1 is overexpressed in pancreatic cancer, and blocking TRPA1 using cannabidiol suppresses tumor growth and reduces metastasis to the lungs (Figure 1). Blocking TRPM8 inhibits pancreatic cancer growth in vitro and in vivo and reduces metastasis to the liver [29]. These findings suggest that TRP channel inhibitors may help treat pancreatic cancer. The research also has shown that pancreatic cancer is a complex disease with multiple risk factors and molecular pathways. The lack of understanding of the pathogenesis of pancreatic cancer, partly due to the difficulty in studying human tissue samples, has impeded the development of effective treatment methods by researchers [10,11,12,13]. While osmotic regulation and pancreatic ductal carcinoma are not directly related, this process is essential for proper functioning cells and tissues, and various mechanisms are involved in regulating it [30]; there may be some indirect connections between the two. For example, changes in osmotic pressure and ion concentrations can affect cellular signaling pathways and may contribute to the development and progression of certain types of cancer [31,32,33,34,35]. Studies have shown that PDAC cells exhibit altered regulation of TRP ion channels involved in osmotic stress response [36,37].

Additionally, the pancreas regulates blood sugar levels and fluid balance in the body, which are related to osmotic regulation [38]. Overall, a potential link between TRP channels and PDAC warrants further discussion. While the relationship to TRP channels remains unclear, it may be pertinent to briefly mention this aspect as a prospective avenue for future research in the review.

**Figure 1 biomedicines-11-01164-f001:**
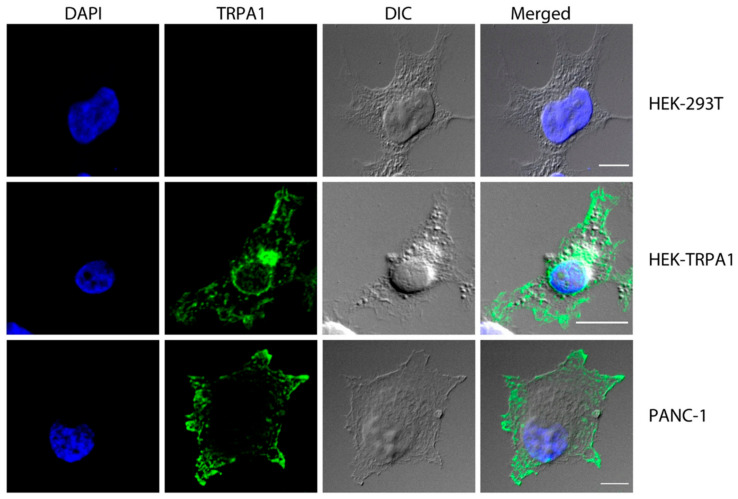
Expression of TRPA1 Protein in PDAC Cells: A. Immunofluorescence of fixed Panc-1 and HEK-293 T cells either transfected or un-transfected with pcDNA3.1 Plasmid encoding human TRPA1. Cells were stained with mouse monoclonal TRPA1 primary antibody (green), whereas DAPI nuclear staining is shown in blue. Scales are included in each image. Bars represent 10 µm. Notes: Reproduced with permission from Ref. [39]. Copyright 2021 Nature Portfolio.

### 3.1. The Distribution of Osmoreceptors

In 1925, British physiologists Starling E. H. and Verney E. B. continuously injected hypertonic saline or sucrose solution into the carotid artery of awake dogs [40]. They found that whenever the hypertonic solution reached the suprachiasmatic nucleus of the hypothalamus, it caused a significant decrease in urine output. With hypophysectomy, this response utterly disappeared. After further research, they hypothesized that the supraspinatus nucleus might be “osmotic receptors” that respond to changes in osmotic pressure in the blood, later confirmed by Ramsay D.J. [40,41].

Furthermore, the subfornical organ (SFO) and the organum vasculum laminae terminalis (OVLT) are the two primary central osmoreceptors located at the anterior wall of the third ventricle in the lamina terminals. In the circumventricular organs (CVO), which lack the normal blood–brain barrier [41,42,43,44,45,46,47], SFO has two distinct types of neurons with opposing actions. A glutamatergic population (SFO-GLUT) promotes thirst and sodium intake, and a GABAergic population (SFO-GABA) inhibits appetite [48]. Peripheral arterial baroreceptors are also a source of signals for specific SFO/OVLT neurons, in addition to their other inputs.

Thus, SFO/OVLT neurons sense plasma osmolality, volume, and blood pressure and then use this information to control thirst. The peripheral receptors in the upper gastrointestinal tract and portal venous system also detect changes in osmolality and blood volume via the TRPV4 receptor [34,36,49,50]. In addition to their role as a central osmolar receptor, they act as a supplementary center for osmoregulation [36].

### 3.2. Osmosensitive Neurons Are Associated with TRPV1

A small increase in extracellular fluid osmotic pressure (greater than 1%) triggers the conversion of this change into electrical signals by hypothalamic osmoreceptors. These signals subsequently stimulate the release of AVP, the sensation of thirst, or both [34,36,50]. AVP and thirst help to reduce salt appetite and induce water intake and reabsorption, leading to a decrease in osmotic pressure [51] (refer to Figure 2).

In recent times, mounting evidence suggests that detecting changes in hypothalamic osmotic pressure is a misano-excitatory coupling process [36,37,38]. Initially, scientists believed these processes were related to a non-selective cation channel (NSCC) sensitive to mechanical stimulation. However, later studies confirmed that a mechanosensitive transient receptor potential vanilloid 1 (TRPV1) channel plays a vital role in the process [36]. The elevated osmotic pressure of extracellular fluid leads to a decrease in cell volume and an increase in the cation conductance of TRPV1. This, in turn, stimulates inward depolarizing current and excitation of OVLT and supraoptic nucleus (SON) neurons located on isotonic ne-sonic pressure, as indicated by sources [36,49,52,53,54,55,56]. On the contrary, hypotonic stimulation of TRPV1 generates hyperpolarizing potentials that inhibit OVLT and SON neurons (Figure 3).

Sorana Ciura has discovered that the OVLT and SON neurons in mice with a TRPV1 knockout (TRPV1^−/−^) exhibit a lack of excitatory responses to hypertonic stimulation. [57]. The response of OVLT neurons to hyperosmotic pressure in wild-type (WT) mice could be blocked by the TRPV1 channel inhibitor ruthenium red (RUR), suggesting that TRPV1 is involved in OVLT sensitivity to hypertonicity [57,58,59].

Fat-soluble substances, such as urea and ethanol, that easily cross cell membranes do not cause changes in cell volume; therefore, they do not stimulate TRPV1 and are called “inactive” osmotic molecules. Only high-concentration sodium chloride and sucrose can form a hypertonic environment in the extracellular fluid, leading the intracellular water to flow out of the cell, thus causing the cell to shrink. Therefore, those changes in cell volume (contraction and swelling) caused by extracellular osmotic pressure are a prerequisite for the activity of TRPV1 [57,58,59].

Further studies have shown that the cation channel expressed on SON permeation-sensitive neurons is not a full-length TRPV1 and may lack motifs required to be sensitive to capsaicin, which specifically ligands to TRPV1 but is an N-terminal variant of the TRPV1channel (ΔN-TRPV1) [36,51,57,58,59]. The N-terminal variant may be involved in coupling changes in cell volume with channel activity or a pore-forming subunit of the transduction channel that facilitates the assembly of elongation-inhibiting channels in osmosensitive cells [36,51,57,58,59].

Ang II can rouse the excitation of magnocellular endocrine cells (MNCs) by activating ΔN-TRPV1, and the excitatory response of MNCs to Ang II disappears in TRPV1^−/−^ mice [60]. The results confirm the role of TRPV1 in osmoregulation. Moreover, RUR-sensitive basal cation conductance is reduced in ADH neurons in TRPV1^−/−^ mice, a phenomenon that can be caused only by TRPV1 acting as part of a pore-forming channel [61]. Thus, the natural osmosensor may be a complex composed of multiple subtypes of TRPV channels. One is SIC, a stretch-inhibited cation channel formed of N-terminal variants of the C-terminal domains of TRPV1 and TRPV4, composed of chimeric channels [58].

Moreover, TRPV 4-mediated Ca^2+^ influx can trigger hypotonicity-mediated activation of hypotonicity-activated anion channels (HAAC) on glial cells, and then glial cells produce taurine or alanine and glycine, and so on [62,63]. Glycine receptor (GlyR) agonists enter osmotically sensitive neurons through GlyR, exerting a robust inhibitory effect on the firing rate of osmotic sensitive neurons, providing a way for the body to detect new hypotonic stimuli explanation [62,63].

### 3.3. ΔThe n-TRPV1 Channel Senses Cell Volume Changes Induced by Changes in Osmotic Pressure

How does ΔN-TRPV1 sense cell volume changes? What is mechanosensitivity mediated by? MNCs are nerve cells with large synapses, which provide a convenient condition for studying this mechanism. Under isotonic conditions, MNCs fire impulses at a slower rate, and this firing activity is inhibited by hypotonicity and enhanced by hypertonicity [60]. The effects of osmotic pressure on MNCs are mediated by many different factors, including the synaptic inputs [64], the release of taurine from local glial cells [65], and the regulation of TRP ion channels expressed by neurosecretory neurons themselves [66].

Studies have shown that osmotic activation of ΔN-TRPV1 occurs in a cytoskeleton-dependent manner in MNCs. This mechanical process is associated with the interaction between the thin layer of actin filaments (F-actin) below the plasma membrane and the densely interwoven network of microtubules (MTs) that occupy most of the cytoplasm of MNCs [51,67,68,69,70]. The involvement of F-actin in the ΔN-TRPV1 activation mechanism remains unclear, but recent findings suggest that MTs interact with ΔN-TRPV1 channels via a C-terminal binding site. This complex plays a role in mediating osmotic sensory transduction. Channel gating requires conductive force. Displacement of this contact prevents channel activation during contraction, whereas increasing the density of these interaction sites facilitates contraction-induced TRPV 1 activation. Consequently, the osmotic sensation is bidirectionally regulated by altering the organization of F-actin and MTs [35,36,51,52,53].

Certain researchers have proposed that the ECM-integrin-TRP channel complex may bind to the microtubule network directly or via actin filaments. When hypertonic cells contract, the cytoskeleton-related network produces a push force on cation channels, facilitating channel activation. In contrast, hypotonic cells swell due to “pulling” forces of the cytoskeleton-related network, which induces the inactivation (hyperpolarization) of the ΔN-TRPV1 channel [67,71]. This model has been proposed by Prager-Khoutorsky M. et al. [53] (Figure 3). The osmotic sensitivity of MNCs cytoskeleton to the ΔN-TRPV1 channel is regulated by phospholipase C (PLC). The PLC isoform, PLCδ1, plays a central role in osmosensory transduction by upregulating F-actin and activating ΔN-TRPV1 channels on MNCs [72]. During MNC action potential firing, PLCδ1 isoforms are activated and exert positive feedback on ΔN-TRPV1 channels to enhance MNC excitability. Angiotensin II (Ang II) can amplify mechanosensitive transduction signals by increasing PLC and protein kinase C (PKC)-dependent F-actin density, enhancing the excitatory response of MNCs to hypertonic stimulation and regulating the release rate of ADH [60,73].

### 3.4. Nax/TRPV 4 Pathway

Studies have shown that significant osmoregulation, such as thirst, still appeared after intraperitoneal injection of hypertonic saline in TRPV1^−/−^ mice. According to the studies, researchers postulate that in addition to TRPV1, other mechanisms can activate OVLT osmotic-sensitive neurons [74,75]. Research has indicated that, without osmotic pressure changes, the increased Na^+^ concentration in cerebrospinal fluid can also cause thirst and the secretion of ADH [48]. An injection of hypertonic NaCl solution in the cerebrospinal fluid of the third ventricle can induce wakefulness in sheep [50,51] and mice [36], with stronger antidiuretic and drinking responses than isotonic hypertonic sucrose solution. These results give evidence that receptors are also sensitive to sodium changes in the brain. Noda M. et al. [76] found that glial cells (astrocytes and ependymal cells) in SFO and OVLT expressed a Na^+^ receptor Nax, which is sensitive to sodium concentration ([Na^+^]) instead of voltage [77]. When the [Na^+^] in the cerebrospinal fluid increases, Nax can be activated. Nax-positive glial cells increase glycolysis to produce lactate, which activates GABAergic neurons [39,78]. “Salt neurons” are AT1a-positive neurons in the SFO that drive salt appetite. Their neural activity is inhibited by these GABAergic neurons, resulting in decreased salt uptake [33]. On the other hand, Nax-positive glial cells are activated to synthesize the endogenous agonist of TRPV4, Epoxyeicosatrienoic acids (EETs), which, in turn, activates TRPV 4 channels (blue) in adjacent neurons to control water uptake (Nax/TRPV 4 pathway) and regulate the water balance in the body [33,64,79].

### 3.5. SLC9A4 Is Sensitive to [Na^+^] Changes

Noda M. et al. showed a marked reduction in water intake after intraventricular (ICV) injection of hypertonic NaCl solution in Nax—knockout mice; nevertheless, there is still a significant amount of water intake, indicating that apart from the Nax/TRPV4 pathway, an unidentified [Na^+^]-dependent pathway also plays a role in water uptake [65]. According to research, SLC9A4, also known as the Na^+^/H^+^ exchanger 4 (NHE4), functions as a [Na^+^] sensor, and SLC9A4 has been found on AT1a-positive neurons (angiotensin receptor two 1a) in the OVLT [39]. Increased [Na^+^] could activate SLC9A4 as well via Ang II, but there is no response to increased osmotic pressure. The knockdown of SLC9A 4 in OVLT reduced the increase in water intake caused by the rise in [Na^+^] in the cerebrospinal fluid, which means activation of SLC9A4 can control water intake [33,64,65,79]. (Figure 4 and Figure 5).

ICV injection experiments with specific inhibitors showed that the increase in extracellular [H^+^] caused by SLC9A4 activation subsequently stimulated acid-sensing channel 1a (AS1C1a) to induce water uptake. Increased sodium concentration ([Na^+^]) in body fluids is implicated in the mechanism by which sympathetic nerve activity (SNA) increases blood pressure (BP) [33,64,65,79].

### 3.6. AngII Regulates the Excitatory Response of MNCs to Hypertonic Stimulation

As mentioned before, AngII can enhance the excitatory response of MNCs to hypertonic stimulation and regulates the release of ADH by increasing PLC and protein kinase C (PKC)-dependent F-actin density on MNCs. AngII neurons also respond to osmotic stimulation by activating epithelial Na^+^ channels (ENaC). Alterations in ENaC activity modulate the firing rate in large cell neurons by causing tonic changes in membrane potential [66]. Alexander Saffran observed that the activation of TRPV1 via capsaicin has a modulatory effect on d-ENaC mRNA and protein expression, as well as ENaC channel function measured as Na^+^ flux in HEK-293 cells [67].

Chronic salt loading leads to the expression of Ca^2+^-permeable GluA1 receptors in newly formed glutamate synapses, and maintaining these synapses requires continuous dendritic protein synthesis. Furthermore, activity-dependent neuropeptide release from neurohypophysis terminals has been observed to exhibit a unique characteristic in recent recordings [66,67,68]. Through voltage-gated Ca^2+^ channels, a significant portion of the voltage-dependent neurohypophysis neurosecretion was Ca^2+^ influx-independent [69]. These discoveries offer significant and fresh advancements in the neuroplasticity and electrophysiological signaling mechanisms of the hypothalamic–neurohypophyseal system, which continue to contribute substantially to neurophysiology.

### 3.7. The Interaction between TRPV1 and BCL2

Previous studies have shown that activation of the heat-sensing ion channel TRPV1 can suppress the growth of cancer cells in a process known as apoptosis [50,58,70,72]. Still, the mechanism by which this occurs remains unclear. In the present study, we sought to identify the role of BCL2 in this process and to elucidate the molecular mechanisms underlying the regulation of the interaction between TRPV1 and BCL2. Recent results revealed that BCL2 inhibits the activity of TRPV1 by promoting its degradation via the ubiquitin-proteasome pathway [71,72,73]. The inhibition of this interaction was mediated by the nuclear receptor PPARγ, which was highly expressed in pancreatic tumors and was associated with poor clinical outcomes [73,80].

## 4. Outcome and Implications

PDAC is one of the most lethal forms of human cancer with a dismal prognosis; the median survival time for patients not surgically treated is less than one year, and only a tiny fraction of patients respond to conventional chemotherapy [1,2,3]. New treatment approaches are, therefore, urgently needed. One type of TRP channel, TRPV1, has been exciting because it senses heat and noxious stimuli in cells [18]. Through interaction with a ligand called capsaicin, TRPV1 also regulates the release of pain-relieving chemicals in the brain, which suggests that it may also play a role in cancer pain. According to studies, the activation of TRPV1 induced by capsaicin has been shown to inhibit cancer cell growth through apoptosis. However, the precise mechanism involved in this process has yet to be fully elucidated. Nevertheless, there appear to be conflicting reports on the specific effects of TRP channel activation and inhibition on PDAC cells. Several studies suggest that TRP channels are essential in oxidative stress-induced cell death [81,82,83]. In particular, the TRPM2 channel has been shown to mediate cell death via downstream mechanisms involving caspase 8, 9, 3, 7, and PARP cleavage [82]. Some studies have reported that specific inhibitors of TRP channels can induce apoptotic cell death in cancer cells, including those in PDAC. In contrast, other studies have suggested activating TRP channels, such as TRPV1, can induce apoptosis in PDAC cells [83,84,85,86].

These contradictory findings may reflect differences in the specific TRP channels targeted by the inhibitors and activators in these studies and differences in the experimental conditions and cell types used. For example, some TRP channels may play a pro-apoptotic role in some cancer cells, while others may be anti-apoptotic or have no effect [87]. Additionally, the effects of TRP channel activation or inhibition on cancer cells may depend on other factors, such as different signaling pathways or differences in the tumor microenvironment. Therefore, further research is needed to clarify the effects of TRP channels on PDAC cells and determine the most effective strategies for targeting these channels in cancer therapy.

Recent studies have uncovered new details about the functional interaction between TRPV1 and the nearby protein BCL2, which protects cells from death in response to stress or injury [11,12,13,14,15]. The aforementioned interaction appears to be modulated by the nuclear receptor known as peroxisome proliferator-activated receptor gamma (PPARγ), which is highly expressed in pancreatic tumors and is associated with unfavorable patient outcomes. These findings suggest that blocking the interaction between TRPV1 and BCL2 may be a promising strategy for inhibiting tumor growth in pancreatic cancer patients. Our brief review article presented these research results and discussed their potential therapeutic implications for the treatment of PDAC [12,13,14,15,16,18,50,58,70,71,72,72,73,80].

Several recent studies have investigated the effect of targeting TRP channels on PDAC growth. Research has found that the inhibition of TRPV4 reduced PDAC cell proliferation and migration and suppressed tumor growth [88,89]. According to a study, TRPM7 plays a partial role in the progressive and invasive nature of PDAC. Silencing TRPM7 in PDAC cells reduces cancer cell invasion. The Hsp90a/uPA/MMP-2 proteolytic axis releases MMP-2, which activates TRPM7, resulting in ECM degradation and promoting the tumor cells’ invasive potential [90].

Similarly, another study by Zhang et al. demonstrated that the inhibition of TRPM8 inhibited PDAC cell growth and migration in vitro and reduced tumor growth in vivo [91]. A study has revealed that TRPM7 plays a crucial role in cellular proliferation in pancreatic epithelia and adenocarcinoma. Silencing did not induce apoptosis but led to replicative senescence with upregulation of p16CDKN2A and WRN mRNA. Combining anti-TRPM7 siRNA and gemcitabine improved cytotoxicity, suggesting that modulation of TRPM7 expression could improve treatment response in pancreatic cancer when combined with apoptosis-inducing agents [92]. The findings of these studies indicate that gemcitabine plays a complex and context-dependent role in PDAC growth and highlight the need for further research to elucidate the mechanisms underlying their effects [92,93].

Tumor therapy has shifted to neoadjuvant-targeted immunotherapy and ion channel therapy. The advances in diagnosis and treatment have improved survival rates for many forms of cancer due to multiple efforts in this area. Nonetheless, despite their strenuous pursuit, researchers have yet to significantly improve survival rates for patients with PDAC. In recent years, considerable data have been accumulated in studying TRP channels. Research on their expression in tumors and potential contributions to various cancers, including PDAC, is diverse. Studies show that these channels are a culprit in developing PDAC and a possible therapeutic target that could support conventional therapies to combat this intractable disease. While many questions remain, each TRP subfamily contributes to this disease, highlighting the need for further studies on the TRP family to uncover additional possibilities. Nonetheless, the researchers have already translated this knowledge effectively on regulating the TRP channels to balance them. They also circumvented these mechanisms to improve treatment efficacy [18,19,20,21,22,49]. TRP channels play an influential role in the development and metastasis of pancreatic cancer. Mechanosensitive TRP channels affect tumor cells and the stroma during various tumor growth processes, establishing them as potential therapeutic targets. They affect pancreatic stellate cells and have roles in cell proliferation, migration, invasion, and death. We also need to focus on a better understanding of the pivotal relationship between these TRP ion channels and the immune system. This relationship is illustrated in the role of TRPC6 in neutrophil chemotaxis [20,21,22]. Consequently, further study is required to unveil the potential roles of TRP channels in developing the tumor microenvironment.

## 5. Future Perspectives

The potential role of TRP channels in PDAC has opened new avenues for developing effective therapies. Notwithstanding, much is still to be learned about this pathway’s underlying biology and potential therapeutic applications. A deeper understanding of TRP channels’ interaction and their signaling pathways in PDAC is required to develop effective therapies. As such, future research should explore the full extent of the TRP channels’ interactome and its role in PDAC development and progression.

While the development of TRP channel inhibitors is in its early stages, some initial studies have shown potential for their use in treating pancreatic cancer. Inhibitors of TRP channels such as GSK2193874 and HC-030031 have shown potential in preclinical studies to treat PDAC. Moreover, combination therapy targeting multiple TRP channels could improve efficacy and overcome drug resistance [6,7,8]. Researchers are now focusing on identifying new inhibitors and understanding how these channels work in cancer. Ultimately, the goal is to develop specific inhibitors for TRP channels that can be used effectively against pancreatic cancer.

Furthermore, one promising area of research is identifying specific TRP channels’ interactome components that drugs or other therapies could target. Identifying TRP channels’ interactome and their downstream signaling pathways is crucial for developing specific inhibitors of these channels. Nevertheless, one of the biggest challenges in developing TRP channels interactome-based therapies for PDAC is the complexity of the pathway and the multiple ways it interacts with other cellular processes. Yet, this complexity also allows researchers to identify novel therapeutic targets and develop more effective treatment strategies. Table 2 summarizes the future research direction of TRP channels interactome as a potential therapeutic target in pancreatic ductal adenocarcinoma, the challenges, and possible solutions.

Given these obstacles, the clinical translation perspectives of TRP channels interactome-based therapies will require a multi-disciplinary approach that integrates basic research, preclinical testing, and clinical trials. Key challenges will include identifying patients most likely to benefit from TRP channels interactome-targeted therapies, developing effective delivery methods, and minimizing off-target effects. However, the potential benefits of such therapies for PDAC patients make these challenges well worth pursuing. By focusing on key research directions, overcoming technical challenges, and translating promising findings into clinical applications, researchers and clinicians can make significant strides in the fight against PDAC.

## 6. Conclusions

PDAC is a highly lethal tumor whose incidence rate has increased steadily in the past few decades [1,2,3]. Moreover, there are no effective treatments, and existing therapies are ineffective because the cancer cells have become resistant to them. Tumor therapy has shifted to neoadjuvant-targeted immunotherapy and ion channel therapy. The advances in diagnosis and treatment have improved survival rates for many forms of cancer due to multiple efforts in this area. Conversely, this is not the case for PDAC, where despite strenuous efforts by researchers, survival is yet to improve in those patients significantly. In recent years, considerable data has been accumulated in studying TRP channels. In particular, the variety of scientific work regarding their expression in tumors and their potential contributions to the development and progression of various cancers, including PDAC. Studies show that these channels are a culprit in developing PDAC, and they are a possible therapeutic target that could support conventional therapies to combat this intractable disease. Further studies are required on the TRP family, as each subfamily is involved in the disease, although much about it remains unanswered and many possibilities may emerge. Targeting TRP channels as a new strategy for treating PDAC shows promising results in preclinical studies. This influence on the development and progression of PDAC is well documented. Undoubtedly, there is a need for further research to comprehensively comprehend the potential pathophysiological functions of TRP channels and how they can be beneficial clinically.

## Figures and Tables

**Figure 2 biomedicines-11-01164-f002:**
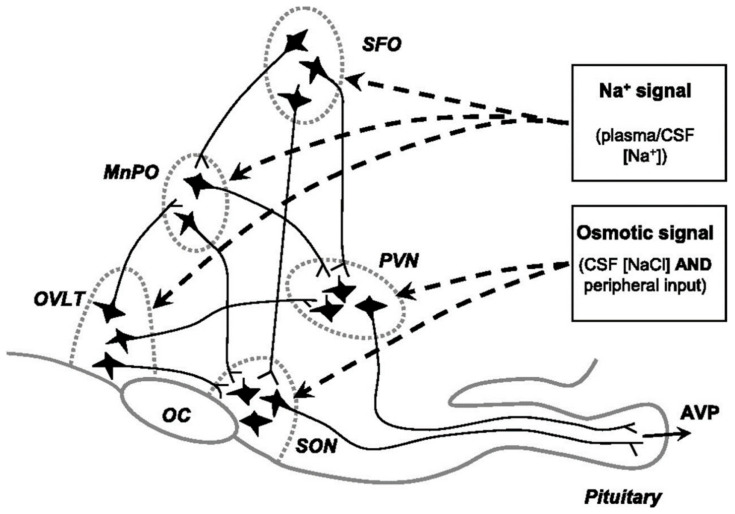
The map of osmoreceptors in the brain: The osmoreceptor’s location in brain regions such as SFO, OVLT, MnPO, and SON. When stimulated by changes in osmotic pressure or sodium levels, the osmoreceptor activates SON, releasing AVP to regulate water reabsorption and stimulate thirst via OVLT and SFO. Na^+^ and osmotic signals are critical factors that regulate osmoreceptor activity and AVP release, maintaining body fluid homeostasis.

**Figure 3 biomedicines-11-01164-f003:**
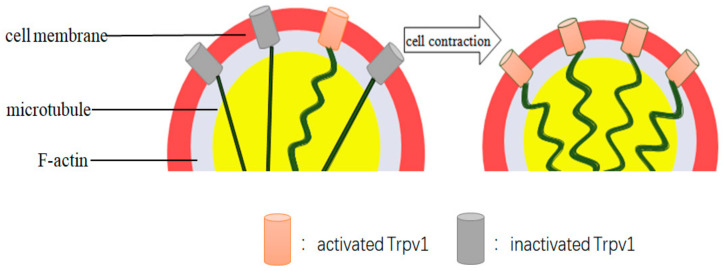
The “push” force-activation model of TRPV 1 is activated by additional microtubule (MT) forces in MNCs. In the resting condition, only a few TRPV 1 channels are activated due to the lack of a corresponding “push” force (left). As hypertonic cells contract, the membrane moves inward, and more TRPV 1 channels are activated by MTS (the right one).

**Figure 4 biomedicines-11-01164-f004:**
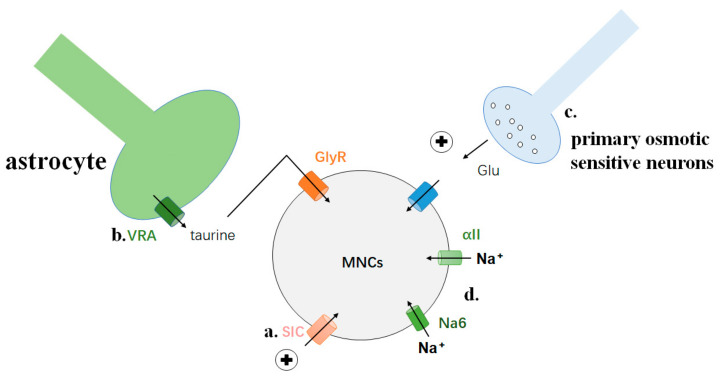
Four unique mechanisms in large cell neurosecretory cells (MNCs) in response to acute osmotic responses. (**a**). Stretch-inhibited channel (SIC) is a stretch-inhibited cation channel on the MEMBRANE of MNCs, also a KIND of NSCC. It transforms the change in cell volume caused by acute osmosis into membrane potential and excitability. That is, hypertonic stimulation of cell contraction leads to an increase in the probability of opening SIC channels, resulting in increased excitability of MNCs ). (**b**). Paracrine effects are associated with adjacent astrocytes. Hypotonic stimulation induces taurine release from adjacent astrocytes through volume-sensitive anionic channels in a permeation-dependent manner. Taurine acts on the glycine receptors of MNCs and inhibits MNCs’ discharge. (**c**). MNCs receive synaptic projections from primary osmotic receptors. (**d**). Additionally, non-inactivated Na^+^ channels formed by the Na6α- subunit may amplify voltage changes induced by other mechanisms. Na^+^ channels encoded by α ⅱ subunit may form channels responsible for action potentials.

**Figure 5 biomedicines-11-01164-f005:**
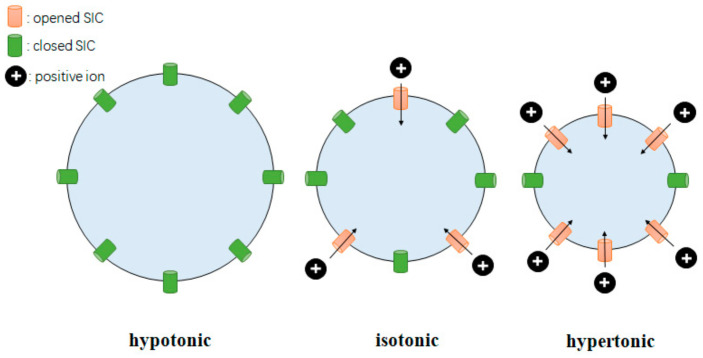
MNCs SIC channel and permeability sensitivity. The diagram shows that hypotonic swelling decreases the probability of SIC channel opening, whereas hypertonic contraction leads to an increase in the same parameters.

**Table 1 biomedicines-11-01164-t001:** Recent studies have shown a role for TRP channels in PDAC.

Reference	Author and Year	Key Findings
[19]	Fallah H. P., et al., 2022	TRP proteins are a large group of ion channels that control many physiological functions in the body and are considered potential therapeutic targets for various diseases, including cancers.
[20]	Chelaru N. R., et al., 2022	Significantly higher expression levels of TRPA1, TRPM8, and TCAF1/F2 were found in tumoral tissues compared to normal tissues, but lower expression levels of TRPV6. The TRP channels have either tumor-suppressive or oncogenic roles.
[21]	Li L., et al., 2022	Research has revealed altered expression of various TRP proteins in numerous cancer types, including PDAC. TRP ion channels are crucial in tumor formation, proliferation, and migration. TRP channel family members have been reported as good prognostic markers and potential targets for cancer drug therapy.
[6]	Mesquita G., et al., 2021	Collected data indicating the TRP family has a potential role in the development and progression of PDAC. It has been found to affect both cancer and pancreatic stellate cells, impacting cell proliferation, migration, invasion, and death. The TRP family may offer new treatments and diagnostics tools for PDAC.

**Table 2 biomedicines-11-01164-t002:** Summary of the future research direction of TRP channels interactome as a potential therapeutic target in pancreatic ductal adenocarcinoma, the challenges, and possible solutions.

Research Direction	Challenges	Possible Solutions	References
Further investigation of TRP channel subfamilies as therapeutic targets	Lack of specificity of TRP channel modulators	Development of more selective TRP channel modulators	[84,94,95]
Evaluation of the role of TRP channels in pancreatic cancer progression and metastasis	Limited understanding of the mechanisms underlying TRP channel involvement in cancer	Use of genetic and pharmacological approaches to elucidate the signaling pathways involving TRP channels	[94,96]
Identification of TRP channel interactors in pancreatic cancer cells	Limited knowledge of TRP channel interactome in pancreatic cancer	Use of proteomic approaches to identify novel TRP channel interactors in pancreatic cancer cells	[94,97]
Development of TRP channel-targeted therapies for pancreatic cancer	Lack of clinical trials and approved TRP channel-targeted treatments for pancreatic cancer	Conducting preclinical studies and clinical trials to evaluate the efficacy and safety of TRP channel-targeted therapies in pancreatic cancer	[89,94]

## Data Availability

The datasets used and/or analyzed during this study are available from the corresponding author upon reasonable request.

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
