# Peer review of "The Emergence of TRP Channels Interactome as a Potential Therapeutic Target in Pancreatic Ductal Adenocarcinoma"

_biomedicines, 2023, doi:10.3390/biomedicines11041164_

Round 1

Reviewer 1 Report

The review article by Wei et. al. on the potential of TRP channels as a novel target for treating pancreatic cancer is interesting and will be of use to the field.  However, there are several major concerns that will need to be addressed.

Major Concerns

1.       The sections on TRP channels really does not relate well to how they might be targets for PDAC, this should be addressed.  In particular, sections 3.2, 3.3, 3.4, 3.5 and 3.6 also section 3.1.  There are no comments on how, if at all, osmotic regulation is related to PDAC.  This whole sections seems irrelevant to this review.

2.       There are several areas that seem to repeat themselves, often with slightly contradictory information.  For example there are 3 references (references 1, 2, & 8) referred to in different parts of the manuscript that refer to the survival rate for pancreatic cancer at 9, 10, and 20%.  These should be combined into 1 statement. 

3.       In line 86 the authors write that specific inhibitors of TRP channels induce apoptosis and cell death in PDAC cells, however in the discussion (lines 353-365) talk about activation of TRP channels (TRPV1) inducing apoptosis in PDAC.  These contradictory statements need to be addressed.

4.       Discussion of more studies targeting TRP channels on PDAC growth would be interesting to see, if available. 

5.       Overall, the manuscript does not read well and is often repetitive.  Major English revision is required. 

 Minor Concerns

1.       In the introduction on line 45 “It is the most common. . . does not appear to belong in this sentence. 

2.       Lines 108-118 are difficult to understand, and it is unclear how this whole section fits with PDAC, since it seems to switch to talking about TRP channels in general and not specifically PDAC.

3.       Lines 190-197 also appear to largely repeat what was discussed earlier in the review.   

Author Response

Dear Reviewer,

Thank you for your valuable feedback on our paper. Your time and effort are greatly appreciated; your comments have helped us improve our work's quality.

We are grateful for your expertise and have incorporated your suggestions into the manuscript.

Best regards

Reviewer 2 Report

This work seems from the title to deal with TRP Channels Interactome as a Potential Therapeutic Target in Pancreatic Ductal Adenocarcinoma (this is the title of this manuscript). Actually, the specific portion dealing with PDAC and this TRP channels is really limited.

Indeed, the paragraph 3 entitled: TRP Channels in Pancreatic Ductal Adenocarcinoma (PDAC), considers a table with some information on these channels in PDAC, but the large portion of the other subparagraphs are not on the topic. Actually, I am not an expert of TRP channels, but I know very well PDAC. I would say that the manuscript is mainly onTRP channels not on these channels in PDAC.

Furthermore, the references indicated in the table seem to do not correspond to the reference list. Indeed, the reference 14 is the follow: Li, H. (2017). TRP channel classification. Transient Receptor Potential Canonical Channels and Brain Diseases, 1-8. Unfortunately, in the table it is indicated as Li, L., Xiao., et al. {2022}. I think there is a mistake. The same for the others, some right references are present at the end of the manuscript (see reference 80 for instance).

However, besides the low attention in preparing the table, the authors do not discuss diffusely the works that are on the topic, but simply insert a sentence in the table. To state that the TRP channels can be a target for therapy of the PDAC the authors should support this with a strong discussion on the topic.

Finally, the reference 82 can be considered a good example of the relevance of TRP channels in PDAC and this present manuscript adds a little to this review.

Author Response

(The authors gave the same response as above.)

Reviewer 3 Report

The Review article titled "The Emergence of TRP Channels Interactome as a Potential Therapeutic Target in Pancreatic Ductal Adenocarcinoma", authored by Yuanyuan Wei and colleagues highlights the significance of Integral membrane proteins known as Transient Receptor Potential (TRP) channels in PDAC.

The review efficiently discusses the challenges associated with PDAC, which is the most aggressive form of pancreatic cancer and TRP as potential therapeutic interventions. The text is informative and well-structured, presenting current knowledge of the molecular role of TRP channels in the development and progression of pancreatic ductal carcinoma. The inclusion of specific examples and the potential of TRP channels as a target for therapeutic interventions, making this a valuable read for scientists in the field.

I congratulate the authors for their effort.

I have the following reccomendations:

Minor revisions:

1) Figure 2 legend should explain the role of osmoreceptor with an extended, self-sustained text, including abbreviations.

2) The text contains some mistakes, including structural errors.

For example:

Line 64 chronic inflammation, oxidative stress, and other environmental factors -> chronic inflammation and oxidative stress are not environmental factors

Line 124 The expression of various TRP proteins has been  altered, and they play a crucial... -> consider revising.

Author Response

(The authors gave the same response as above.)

Round 2

Reviewer 1 Report

The revised manuscript is much improved and addresses all of my original concerns.  There are still a few areas that are difficult to read, but overall the English is markedly improved from the first submission.

Author Response

Dear Reviewer,
Thank you for getting back to me regarding the revised manuscript. I am pleased to hear that the manuscript has addressed all your initial concerns and that the English has markedly improved from the first submission.
We apologize for any remaining areas that may be difficult to read, and we have reviewed the manuscript thoroughly to identify and address any remaining issues.
Thank you again for your assistance in improving the quality of the manuscript.
Best regards,
AK Taha
MD/Ph.D.
Professor and High-End Foreign Expert
Auckland, New Zealand
MMC Reg. No. 74494
ORCID: https://orcid.org/0000-0003-3831-2459
Scopus Author ID: 57201420645
Google Scholar: AT Khalaf
Web of Science ResearcherID: AAN-4913-202

Reviewer 2 Report

The authors have made some improvements and the manuscript is more informative in this revised version.

Author Response

Dear Reviewer,
Thank you for taking the time to review the revised manuscript. We are pleased to hear that our efforts to improve the manuscript have resulted in a more informative version.
We have thoroughly reviewed the manuscript and implemented changes where necessary to enhance its overall quality and readability.
Thank you again for your feedback and valuable input.
Best regards,
Best regards,
AK Taha
MD/Ph.D.
Professor and High-End Foreign Expert
Auckland, New Zealand
MMC Reg. No. 74494
ORCID: https://orcid.org/0000-0003-3831-2459
Scopus Author ID: 57201420645
Google Scholar: AT Khalaf
Web of Science ResearcherID: AAN-4913-202
